# Elimination of oxygen sensitivity in $\alpha$-titanium by substitutional alloying with Al

Yan Chong[1,2,5], Ruopeng Zhang [1,2,5], Mohammad S. Hooshmand[1], Shiteng Zhao[1,2,4], Daryl C. Chrzan[1,3], Mark Asta [1,3], J. W. Morris Jr [1] & Andrew M. Minor [1,2,3✉]

Individually, increasing the concentration of either oxygen or aluminum has a deleterious effect on the ductility of titanium alloys. For example, extremely small amounts of interstitial oxygen can severely deteriorate the tensile ductility of titanium, particularly at cryogenic temperatures. Likewise, substitutional aluminum will decrease the ductility of titanium at low-oxygen concentrations. Here, we demonstrate that, counter-intuitively, significant additions of both Al and O substantially improves both strength and ductility, with a 6-fold increase in ductility for a Ti-6Al-0.3 O alloy as compared to a Ti-0.3 O alloy. The Al and O solutes act together to increase and sustain a high strain-hardening rate by modifying the planar slip that predominates into a delocalized, three-dimensional dislocation pattern. The mechanism can be attributed to decreasing stacking fault energy by Al, modification of the "shuffle" mechanism of oxygen-dislocation interaction by the repulsive Al-O interaction in Ti, and micro-segregation of Al and O by the same cause.

[1] Department of Materials Science and Engineering, University of California, Berkeley, CA, USA. [2] National Center for Electron Microscopy, Molecular Foundry, Lawrence Berkeley National Laboratory, Berkeley, CA, USA. [3] Materials Sciences Division, Lawrence Berkeley National Laboratory, Berkeley, CA, USA. [4] Present address: School of Materials Science and Engineering, Beihang University, Beijing 100191, China. [5] These authors contributed equally: Yan Chong, Ruopeng Zhang. ✉email: aminor@berkeley.edu

Solution hardening is one of the most effective ways to tailor mechanical properties of structural materials, as alloying with either interstitial or substitutional atoms creates internal stress fields that impede dislocation motion[1]. Despite the successful application of solid solution strengthening in a wide range of metal alloys, anomalous alloying sensitivity can occur when intense chemical interactions exist among the solute atoms and the matrix atoms[2–5]. Such interactions lead to extreme sensitivity to solute concentrations that can decrease instead of increase both strength and ductility of the alloys[4,6–8]. Understanding these sensitive interactions involving interstitial and substitutional atoms is key to either harnessing or mitigating the potent effects to achieve the desired synergy of strength and ductility.

Hexagonal close-packed (HCP) $\alpha$-Ti-based alloys are particularly attractive systems owing to their technological importance in weight- and corrosion-sensitive environments[9,10]. Commercially pure $\alpha$-Ti deforms in tension through a combination of deformation twinning and dislocation glide[9,10]. It has excellent ductility at cryogenic temperatures as low as 77 K, but rather low strength[11] and, hence, is usually alloyed to achieve desirable mechanical properties. Interstitial oxygen is known to be a potent hardening agent for Ti and has two predominant effects: it inhibits twinning[11–14], and it impedes dislocation glide through direct solute-dislocation interactions[4,11]. However, the specific solute hardening mechanism has the disadvantage that it promotes a planar slip pattern that causes a dramatic loss of ductility. As recent research has shown[11], dislocation glide past an O interstitial can cause its displacement from its stable octahedral site to a hexahedral site where it offers less resistance to dislocation glide. As a consequence, a region that has been swept by a dislocation is on average softened, promoting concentrated planar slip. Since the recovery of the O interstitial distribution requires diffusion, planar slip and the associated loss of ductility become more pronounced as temperature decreases. Hence, as we show below, Ti-0.3 O has high strength, but almost no elongation at 77 K[11]. Compared with a strain-field interaction via solid solution strengthening, the shuffling mechanism will mechanically shuffle the crystallographic sites of oxygen atoms, leading to strong oxygen sensitivity as described in our prior work[11].

Substitutional aluminum is more commonly used to strengthen Ti. Al additions also inhibit twinning[15] and introduce obstacles to dislocation glide. In this case, the obstacles are short-range ordered (SRO) domains in a $DO_{19}$ ($Ti_3Al$) pattern (particularly above 6.0 wt.% Al)[16–20]. These obstacles also induce planar slip at ambient temperature with some loss of ductility. The well-established mechanism, in this case, is the relatively low-stress propagation of pairs of dislocations in which the leading dislocation creates an anti-phase boundary in the ordered domain. The trailing dislocation partially restores the order within the SRO domain and, hence glides more easily[16–20]. Successive pairs gliding on the same plane increasingly shear the ordered domain, weakening it and potentially dissolving it. However, while a significant Al addition to Ti does produce a deformation pattern that is conducive to good ductility even at cryogenic temperature, it does not impart strength competitive to that achievable with rather small oxygen additions. This raises the question of whether the beneficial features of the two alloy additions can be combined to produce Ti-Al-O alloys that offer the low-temperature strength of Ti-O along with the ductility of Ti-Al. This possibility is particularly intriguing given recent work by Gunda, et al.[21], whose DFT calculations show a strong Al-O repulsive interaction in Ti-Al-O alloys. Owing to this strong nearest-neighbor repulsion, oxygen atoms tend to occupy octahedral sites that are exclusively coordinated by Ti atoms. This suggests the possibility of chemical inhomogeneity that may disrupt the oxygen-induced planar slip.

In this study, we experimentally demonstrate that the synergy of 6 wt.% Al and 0.3 wt.% O does produce an excellent combination of high strength (~1.3 GPa) and good ductility (~25%) at cryogenic temperature. The metastable mechanical shuffling mechanism of oxygen is interrupted by the SRO of aluminum, leading to an order-of-magnitude enhancement of the ductility while maintaining the benefit of the potent strengthening of interstitial elements. The atomistic origin of the transition is discussed in light of transmission electron microscopy (TEM) observations and density functional theory (DFT) calculations. The findings presented here enable the development of low-cost titanium alloys by lifting the rigorous processing required to control interstitial impurities, broadening the spectrum of the materials' engineering applications.

## Results and discussion

**Improved mechanical properties by an aluminum addition.** Figure 1a exhibits the engineering stress–strain curves for Ti and Ti-$x$Al-$y$O alloys ($x = 2$, 4, and 6, $y = 0.1$ and 0.3 wt.%) in liquid nitrogen (~77 K) at a strain rate of $10^{-3}$ s$^{-1}$. The results show that, as reported previously[11], increasing the oxygen content of Ti-$y$O alloys increases strength but dramatically decreases ductility, limiting applications at cryogenic conditions. In the present work 2.0~6.0 wt.% Al was added to Ti-0.1 O and Ti-0.3 O alloys to explore their influence on strength and ductility. The data in Fig. 1a show a monotonic increase in tensile strength with Al content at both oxygen levels. However, there is a dramatic difference in the effect of Al on ductility. In Ti-$x$Al-0.1 O ($x = 0$, 2, 4, and 6) alloys (red lines in Fig. 1a), the increase of strength came with a loss of strain-hardening and, hence, tensile ductility (Fig. 1c), as is the usual case with engineering alloys. But in Ti-$x$Al-0.3 O ($x = 0$, 2, 4, and 6) alloys (blue lines in Fig. 1a), the strain-hardening and tensile ductility were significantly improved by increasing aluminum content (Fig. 1d). It follows that the catastrophic brittle failure of the Ti-0.3 O alloy at cryogenic temperature can be eliminated by the substitutional alloying of aluminum, which simultaneously provides a substantial increase in strength. Looking at the same data from the perspective of oxygen content, the detrimental influence of oxygen on tensile ductility in Ti-O alloys is overcome by the addition of 6.0 wt% aluminum; the total elongation of the high-oxygen alloy Ti-6Al-0.3 O (~25%) is greater than that of Ti-6Al-0.1 O (~22%), and even that of low-O Ti-6Al (~20%). And this improvement in ductility is accompanied by a substantial increase in tensile strength, from 1050 MPa (Ti-6Al) to 1100 MPa (Ti-6Al-0.1 O) to 1260 MPa (Ti-6Al-0.3 O).

The combined effects of aluminum and oxygen contents on the toughness (as measured by the area beneath the stress–strain curve) of the various Ti-O and Ti-Al-O alloys are shown by a 3D-plot in Fig. 1b. The results show that Ti-6Al-0.3 O has comparable tensile toughness to pure Ti and Ti-0.1 O with roughly 3× the tensile strength. In comparison, although Ti-0.3 O and Ti-2Al-0.3 O alloys exhibit high strength at 77 K (~1000 MPa), their toughness at cryogenic temperature is severely limited due to an early fracture. Consistent with this, the cryogenic fracture mode changed from intergranular brittle fracture in Ti-0.3 O to transgranular ductile fracture (predominantly dimpled rupture) in Ti-6Al-0.3 O (Supplementary Figure 1).

**Deformation behavior: the contribution of dislocation activity and deformation twinning.** The deformation of $\alpha$-titanium occurs through a combination of dislocation plasticity and mechanical twinning[9,12,16,22–28]. Mechanical twinning can be particularly important at higher strains since it increases work

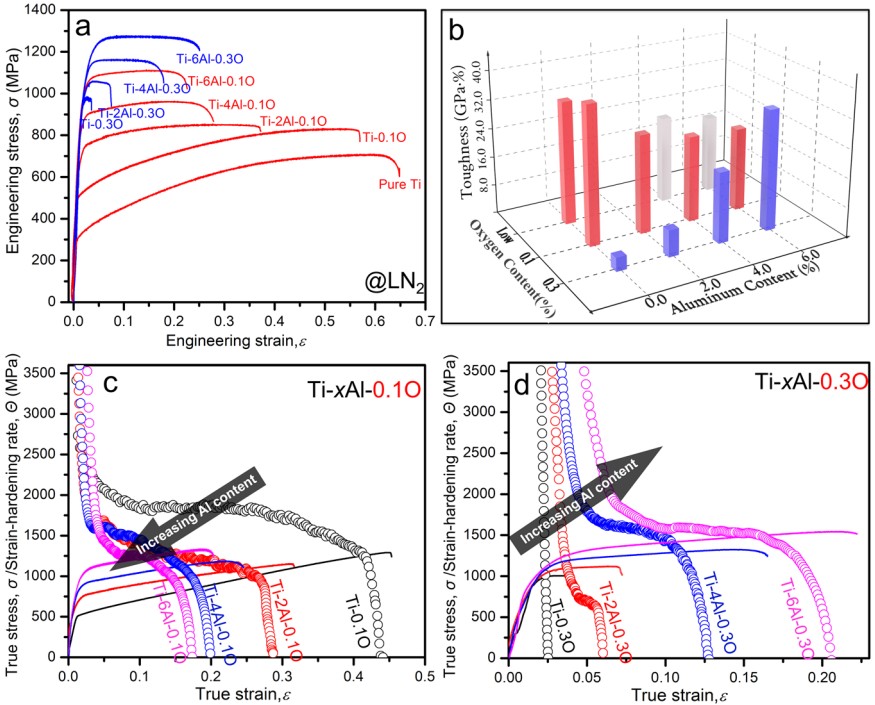

**Fig. 1 Mechanical properties of Ti-Al-O ternary alloys at liquid nitrogen temperature. a** Engineering stress–strain curves of Ti-O and Ti-Al-O alloys tested in liquid nitrogen at a strain rate of $10^{-3}$ s$^{-1}$ (all compositions in wt.%). **b** 3D-plot showing the combined effects of aluminum and oxygen contents on the tensile toughness (the area covered beneath the engineering stress–strain curves) of Ti-O and Ti-Al-O alloys in liquid nitrogen. True stress–strain curves (solid lines) and strain-hardening rate curves (symbols) of Ti-$x$Al-0.1 O **c** and Ti-$x$Al-0.3 O **d** alloy systems ($x = 0$, 2, 4, and 6) in liquid nitrogen. The strain-hardening rates and tensile ductilities of the two alloy systems trend in opposite directions with increasing aluminum content.

hardening by breaking up the microstructure and inhibiting planar slip. Both substitutional (Al) and interstitial (O) solutes influence twinning behavior in $\alpha$-titanium. Al contents above ~4.0 wt% cause a dramatic loss of twinning activity in Ti-Al binary alloys[18]. The effect of oxygen on twinning activity in Ti-O is less clear, although suppression of conventional deformation twins with increasing oxygen content has been reported at room temperature[2,11,13,14]. There does not appear to have been prior work on twinning behaviors in Ti-Al-O alloys. To address this issue, we used electron backscattered diffraction (EBSD) to identify the extent and pattern of mechanical twinning in the Ti-O and Ti-Al-O samples fractured in liquid nitrogen (Fig. 2). The results show that twinning activity is decreased by the increase of both aluminum and oxygen contents. In Ti-$x$Al-0.1 O alloys ($x = 0$, 2, 4, and 6), the extent of twinning decreases with Al content, and is essentially suppressed at an aluminum content of 4.0 wt% and above (Fig. 2). In the higher oxygen Ti-$x$Al-0.3 O alloys ($x = 0$, 2, 4, and 6), twinning is relatively rare in Ti-0.3 O and becomes totally suppressed at an aluminum content of 2.0 wt % and above (Fig. 2).

The kernel average misorientation (KAM) maps of Ti-Al-O alloys after tensile fracture at 77 K are also examined (see Supplementary Discussion and Supplementary Figure 2), in order to qualitatively compare the contributions of grain boundaries and grain interiors to the plastic deformation. It is clearly shown that in the high Al and O content microstructures, the KAM values (indictor of plastic deformation) became more spread inside the grains, rather than localized near the grain boundaries as frequently seen in the low Al and O content counterparts. The more homogenous distribution of plastic deformation in Ti-6Al-0.3 O alloy can be beneficial for the tensile ductility in two ways. First, the substantial dislocation interactions/cross-slips inside the grains can provide sufficient strain-hardening ability to sustain the high-stress level at 77 K (which

will be discussed in detail in the following part). Second, the strain localization at the grain boundaries can be somehow relaxed, preventing/delaying the formation of grain boundary micro-cracks.

Reviewing the cryogenic tensile test data presented in Fig. 1 in light of these results suggests that the strength and ductility of the Ti-$x$Al-0.1 O alloy are strongly influenced by mechanical twinning. As the Al content increases, the twinning component decreases, raising the strength and decreasing the strain-hardening rate, which lowers ductility. The absence of twinning in Ti-6Al-0.1 O shows that its mechanical properties are solely due to dislocation plasticity. Similarly, the limited or absent twinning in the Ti-$x$Al-0.3 O alloy means that its mechanical properties are also governed by dislocation plasticity. In a typical alloy, this would produce an inverse correlation between strength and ductility[29,30]. But the opposite trend appears in Ti-$x$Al-0.3 O: strength, strain-hardening rate, and tensile ductility increase together as the Al content are raised from 0 to 6 wt.%. The pronounced influence of Al on strain-hardening is particularly apparent in the true stress-strain curves given in Fig. 1d. Although the Al addition has very little effect on behavior at very small strains, near yield; it dramatically increases the strain-hardening rate at even moderate strain leading to a very favorable combination of strength and ductility. Interestingly, and in contrast to the behavior of Ti-$x$O, increasing the oxygen content of Ti-6Al-$y$O alloys from $y = 0.1$ to $y = 0.3$ also improves the strain-hardening rate, leading to higher strength and greater tensile ductility. A synergy of 6.0 wt% aluminum and 0.3 wt% oxygen provides the best combination of strength and ductility at cryogenic temperature.

The most likely cause of improved strain-hardening behavior in dislocation plasticity is a favorable change in the pattern of the dislocation network that develops during straining. To investigate that possibility, we selected three alloys; Ti-2Al-0.3 O, Ti-6Al-

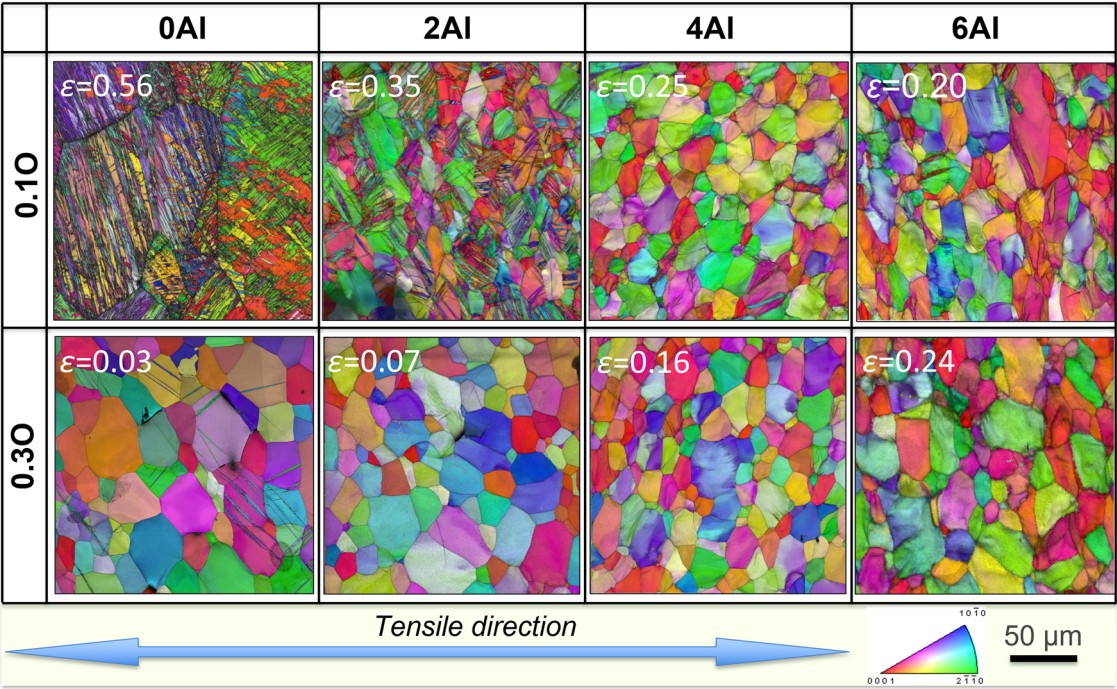

**Fig. 2 EBSD inverse pole figure (IPF)+ image quality (IQ) maps of Ti-O and Ti-Al-O alloys after tensile fracture in liquid nitrogen.** The fracture strain was also indicated in each microstructure. When the oxygen content is 0.1 wt%, the deformation twinning was basically suppressed with 4wt.% of aluminum addition, whereas when the oxygen content is 0.3 wt%, the deformation twinning was basically suppressed with 2wt.% of aluminum addition. The scale bar applies to all panels in this figure.

0.1 O, and Ti-6Al-0.3 O for detailed analysis via TEM. Twinning is essentially suppressed in these alloys (Fig. 2), whereas substantial differences in the strain-hardening rate and tensile ductility are still observed (as shown by the strain-hardening rate curves in Fig. 3k).

The TEM images presented in Fig. 3 show typical dislocation morphologies in the three alloys at interrupted cryogenic plastic strains (2.0%, 6.0%, and 10.0%) as well as after tensile fracture. The corresponding true stress-strain and strain-hardening rate curves (Fig. 3k) are also shown for reference. At the early stage of deformation (2.0%), planar slip bands were observed in all three alloys (Fig. 3a–c), which could be attributed to either mechanical shuffling of oxygen solutes (in the Ti-2Al-0.3 O alloy)[11] or the disruption of short-range ordered aluminum (in the Ti-6Al-0.1 O alloy)[19], or possibly the combination of both (in the Ti-6Al-0.3 O alloy). These planar slip bands constituted a microscopically localized deformation that reduces strain-hardening ability[2,11,12,15]. At a larger deformation strain (6.0% for Ti-6Al-0.1 O and Ti-6Al-0.3 O, 7.0% for Ti-2Al-0.3 O), however, the dislocation morphologies of the three alloys differed significantly. In the Ti-2Al-0.3 O alloy (Fig. 3f), the planar slip bands became even more concentrated on two sets of prismatic planes that intersect at an angle of ~120°, which greatly hindered dislocation cross-slip and interaction, diminishing strain-hardening. As a result, the strain-hardening rate of the Ti-2Al-0.3 O alloy quickly decayed to zero, and the alloy fractured with limited total elongation (~7%). Interestingly, this pattern also developed in Ti-0.3 O alloy where planar slip was caused by the mechanical shuffling of oxygen solutes, and also led to a limited strain-hardening with low elongation to fracture.

However, the dislocation distribution in Ti-6Al-0.1 O alloy evolved a qualitatively different morphology at an intermediate strain (Fig. 3e). In addition to the pre-existing planar slip bands, considerable non-planar dislocation slip was also observed in the interstices between the bands. A typical example of this dislocation arrangement was investigated in detail with the

results shown in Fig. 4. Figure 4a shows two planar slip bands with non-planar dislocations in between as observed from a zone axis of [2–1–10] (a crystal schematic illustration is given in Fig.4f). These dislocations were confirmed to be <**a**> type dislocations according to the diffraction contrast analysis (Fig. 4b). Weak beam dark-field imaging reveals further details of the dislocation configurations (Fig. 4c–e). Inside the planar slip bands that formed in the early stage of deformation, there were numerous dislocation tangles in place of the well-aligned dislocation pileups previously observed (see Fig. 4e). We speculate that the jagged dislocations morphologies, the intensive dislocation tangles, and the large dislocation looping out of the slip bands indicate the occurrence of cross-slip events inside the planar slip bands. More importantly, the nodes of dislocation tangles are consistent with the theory that they can act as Frank-Read sources to emit dislocation loops into the interstitial volumes between these slip bands, and many of these emitted dislocations had evident cross-slip (as indicated by blue arrows in Fig. 4d and e). Thus, the plastic deformation that was initially localized inside the planar slip bands was progressively transferred into the relatively less deformed interstitial volumes, leading to a "deformation delocalization" phenomenon. At a macroscopic level, the deformation delocalization, as well as the enhanced dislocation cross-slip, retarded the decay of the strain-hardening rate, as evidenced by a small plateau stage in the strain-hardening rate curve of Ti-6Al-0.1 O alloy (Fig. 3k). With a further increase in the plastic strain (10.0%), the deformation delocalization in the Ti-6Al-0.1 O alloy continued and the dislocations became more evenly distributed, although the original planar slip bands can still be seen (Fig. 3h). After a tensile fracture, a high density of dislocations in the form of dislocation forests appeared throughout the microstructure (Fig. 3j). Considering the low-oxygen content (0.1 wt%), the dislocation dynamics observed in the Ti-6Al-0.1 O alloy were representative of cases in which the formation of planar slip was

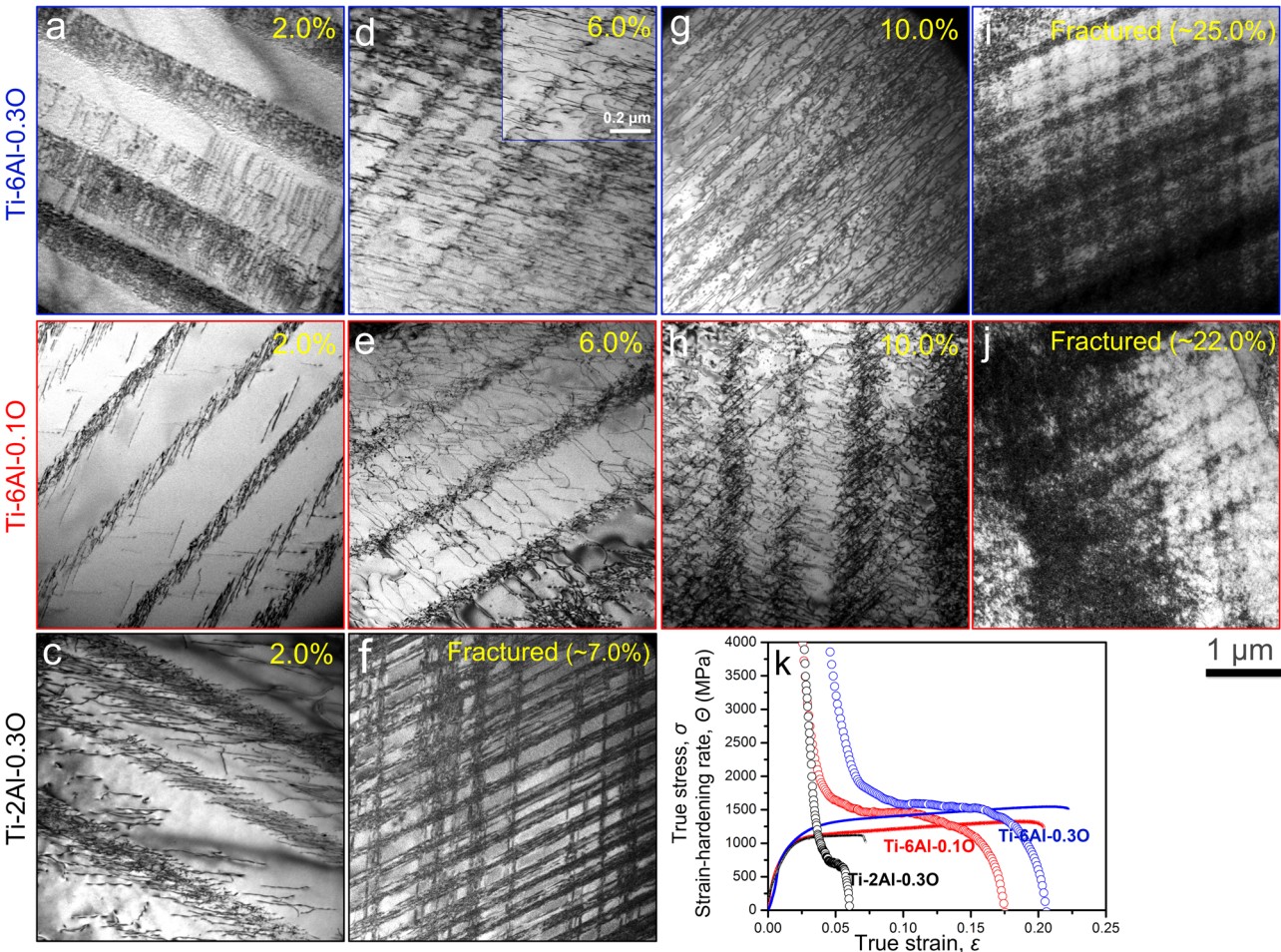

**Fig. 3 Typical dislocation morphologies of Ti-2Al-0.3 O, Ti-6Al-0.1 O, and Ti-6Al-0.3 O alloys after tensile deformation at a cryogenic temperature at selected strains and eventual fracture.** The corresponding strain-hardening rate curves of the three alloys (**k**) are also given for reference. At the early stage of deformation (2.0%) **a**–**c** planar slip bands were observed in all three alloys. With further increase of deformation (from 6% to fracture), the localized deformation became even severe in Ti-2Al-0.3 O alloy **f**, whereas in Ti-6Al-0.1 O and Ti-6Al-0.3 O alloys, a clear delocalization of deformation occurred as dislocations entanglements and possible cross-slips were frequently observed in between the planar slip bands **e**, **h**, **j** and **d**, **g**, **i**. In Ti-6Al-0.3 O alloy, the delocalization occurred even earlier than in Ti-6Al-0.1 O alloy. The scale bar applies to all panels in this figure.

due to the disruption of short-range ordered aluminum solutes. Unlike the ever-exacerbating deformation localization in Ti-2Al-0.3 O alloy where the planar slip was caused by the mechanical shuffling of oxygen solutes, the SRO-induced planar slip exhibited an intrinsic deformation delocalization ability by allowing dislocation interaction and cross-slips in between slip bands at a larger deformation strain. This ability greatly helped to preserve the strain-hardening rate and prevent early fracture of the material.

The evolving dislocation pattern observed in Ti-6Al-0.1 O was repeated and exaggerated in Ti-6Al-0.3 O, consistent with its superior strain-hardening. The initial slip bands predominated after 2% strain in this alloy as well but were denser and less regular in structure than those in Ti-6Al-0.1 O. By 6% strain a significant density of dislocations had been emitted into the interstitial volumes, presumably by the same mechanisms that were operative in Ti-6Al-0.1 O (Fig. 4). The contrast of the original planar slip bands became even weaker at the plastic strain of 6.0% (Fig. 3d), indicating a more homogeneous deformation behavior. As shown inset in Fig. 3d, cross-slip between curved dislocations were frequently observed. With further increase of the plastic strain, a large number density of wavy dislocations was observed throughout the microstructure (Fig. 3g), which eventually led to the formation of dislocation forests at the end

of the tensile deformation (Fig. 3i). By 10% strain, the dislocation distribution was approaching three-dimensional uniformity, though the original slip bands can still be seen. There is an immediate correlation between the enhanced strain delocalization in Ti-6Al-0.3 O and its superior combination of strength and toughness.

## Discussion

The results reported above show that, contrary to expectation, high-O T-Al alloys can have exceptional tensile properties at low temperatures. In particular, Ti-6Al-0.3 O achieves a tensile strength of 1.26 GPa with total elongation of 25% at 77 K. A principal source of these good properties is a high and sustained strain-hardening rate that is owing to the delocalization of planar slip into a relatively diffuse, three-dimensional dislocation pattern as plastic strain increases. This behavior reflects a surprising synergy between Al and O solutes in Ti. Tensile properties improve if Al is added to Ti-0.3 O, an alloy that is brittle at 77 K. Tensile properties also improve if O is added to Ti-6Al, an alloy that is ductile at 77 K, but significantly lower in both strength and ductility than Ti-6Al-0.3 O. We now consider the influence of Al and O, and how they may have a synergy in their contributions to cryogenic tensile properties.

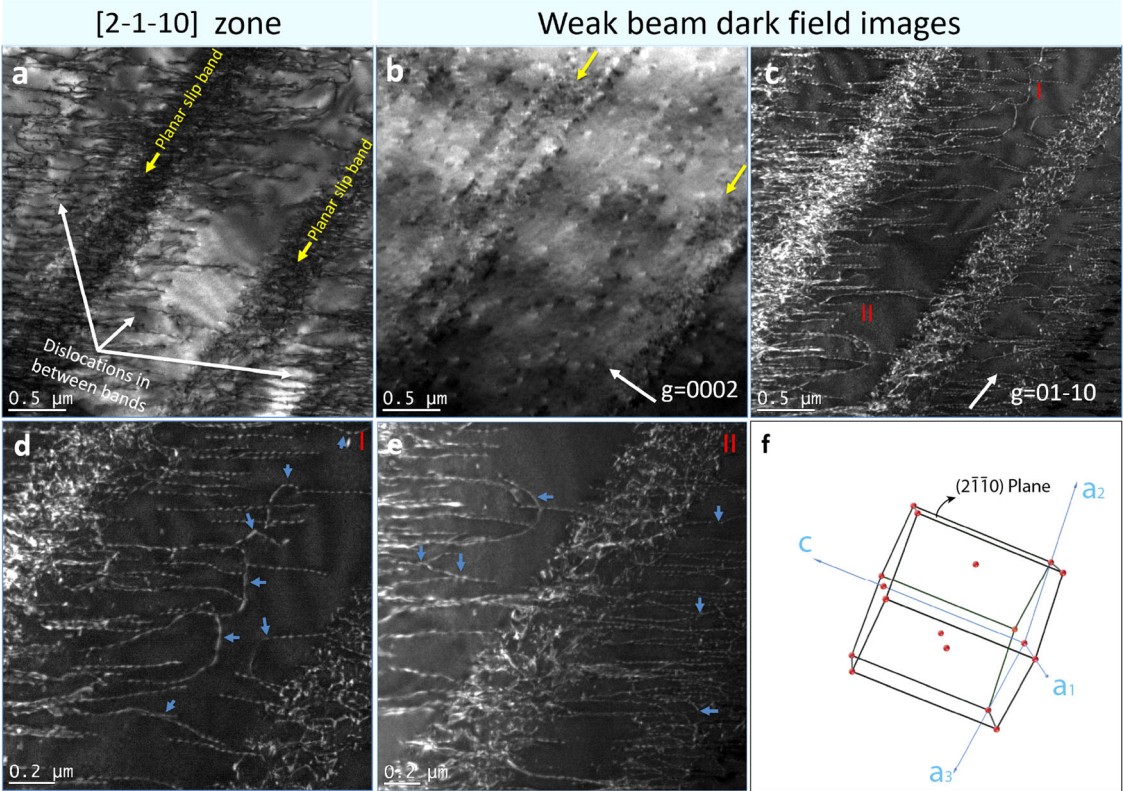

**Fig. 4 Detailed analysis of dislocation morphologies in Ti-6Al-0.1 O after tensile deformation to 6.0% strain at cryogenic temperature. a** Two planar slip bands and some scatteredly distributed dislocations in between the bands were observed from the zone axis of [2-1-10]. **b–e** Weak beam dark field (WBDF) images of the same area. **b** Both planar slip bands and dislocations in between the bands disappeared when $g = 0002$, which excluded the existence of $<c + a>$ dislocations. **c** WBDF image using $g = 01$–10, highlighted the dislocation loops nucleated from the slip bands, as well as the frequent occurrence of possible dislocation cross-slips in between the bands. Some typical examples (areas I and II) were indicated by blue arrows in **d** and **e**. A schematic of the crystallographic orientation for the imaging direction is given in **f**.

To begin with Al, Ti-6Al alloys show significant ductility at 77 K, in fact, better ductility than at ambient temperature. There appear to be two factors that are relevant to this behavior. The first concerns the mechanism of planar slip that compromises strain-hardening and ductility at ambient temperature. Al solutes in Ti tend to associate into (SRO) clusters based on the $DO_{19}$ ($Ti_3Al$) structure. Dislocations preferentially cut through these clusters in pairs, with the lead dislocation creating an anti-phase boundary that is removed by the trailing one, promoting planar slip. However, successive dislocations accumulate deformation that gradually disperses SRO. At ambient temperature local diffusion can restore SRO, maintaining planar slip. However, at cryogenic temperature, the rate of diffusion is insufficient to restore SRO. The resulting, diffuse SRO imposes only a modest energy barrier for cross-slip. With increased dislocation density in slip bands at a larger plastic strain (~6.0%), it is possible for the dislocations to cross slip and avoid repulsion caused by pileups[19]. This trend is confirmed in the current study as seen in Fig. 4. The nodes of dislocation cross-slips would then act as Frank-Read sources to emit dislocations into the undeformed areas in between the slip bands, facilitating deformation delocalization in a microscopic way.

However, this mechanism does not offer a complete explanation. As shown in this work, strain delocalization is more pronounced at higher Al contents. A simple model would predict the opposite since higher Al contents should create SRO domains that are better defined and more easily restored. The second relevant factor, which seems to resolve this issue, is enhanced basal slip at higher Al content[9,18,27] owing to a decrease in the

difference between the critical resolved shear stresses for basal and prismatic slip. In fact, a certain amount of basal dislocation slip aligned perpendicular to the prismatic planar slip bands was present in Ti-6Al-0.1 O at a plastic strain of 6.0%. The easy activation of basal slip as a supplement to prismatic slip contributes to the diffusion of the dislocation distribution to create a more homogeneous plastic deformation. The enhanced basal slip in Ti-6Al-0.1 O alloy compared with Ti-2Al-0.3 O alloy is supported by first-principles DFT calculations (Supplementary Figure 3). These calculations addressed the effect of Al on stacking fault energy (SFE), since SFE plays an important role in dislocation behavior. In particular, for $<a>$-type screw dislocations in Ti the energy value governing 1/3 $<10$-$10>$ slip, called the $I_2$ intrinsic stacking fault (ISF) energy, determines the tendency of the dislocation to dissociate on the basal planes. Our DFT calculations on permutations of a special quasi-random structure (SQS) Ti with 6.0 wt.% (10.0 at.%) Al revealed that the addition of Al reduces the ISF energy compared with pure Ti (Supplementary Figure 3). This finding suggested that Al promotes the glide of the $<a>$-type screw dislocations on the basal rather than the prismatic planes. We also identified variations in SFEs for different local chemical compositions within and around the fault planes. Supplementary Figure 3a shows an SQS cell corresponding to Ti-10at %Al sample. Each SQS contained multiple slip planes with distinct atomic arrangements. Local variation of SFEs as a function of the slip plane of choice for this SQS cell is shown in Supplementary Figure 3b. Although in all cases SFE is reduced compared with the bulk Ti, there is a variation in SFE which depends on the local chemical bonding between atoms (and its change due

to the slip) and concentration of solute atoms at the fault plane. Our findings confirm that Al solutes can promote <a>-type screw dissociation on the basal planes by decreasing the $I_2$ ISF energy.

The significant improvement in tensile properties of Ti-6Al-0.3 O over those of Ti-6Al-0.1 O shows that oxygen also enhances the strain-hardening ability in these alloys, despite its negative influence on the Ti-0.3 O binary. Moreover, comparing the TEM micrographs of the dislocations structures in the two alloys shows that it does this by enhancing the delocalization of dislocation slip. Consideration of the possible effect of the oxygen interstitial solutes led us to identify three possible factors: solute hardening by O, enhanced solute hardening due to the repulsive interaction between Al and O in Ti, and chemical segregation induced by Al-O repulsion.

First, assuming both Al and O in solution, a high Al content, and planar slip initiated by the Al SRO, there will be O interstitials dispersed through the interior of the active slip bands. These will act as obstacles to the dislocations in the band such that higher stress is required to continue planar slip. This higher stress increases the stress available to drive cross-slip or slip onto alternate (e.g., basal) slip planes, the processes that cause dislocations to be emitted from the slip band into the interband spaces, delocalizing the slip.

Second, DFT calculations[21] have revealed an apparently strong repulsion between Al and O atoms in Ti matrices. Even if both are in a roughly homogeneous solution there should be a strong tendency for O interstitials to occupy sites that are surrounded by Ti atoms only. The interaction between an O interstitial and a gliding <a>-type screw dislocation causes a displacement ("shuffle") of the oxygen interstitial from the stable octahedral site to a metastable hexahedral site that offers less resistance to subsequent dislocations. This is the mechanism that promotes planar slip in Ti-O[11]. The shuffle will be impeded if the hexahedral site has Al neighbors, hence increasing the resistance to dislocation glide. Moreover, the higher energy of the hexahedral site with Al neighbors will decrease the activation energy for the return of the interstitial to its original octahedral site, removing the impetus for planar slip. This picture is supported by DFT calculations presented in Fig. S4, as discussed in detail in the supplementary text. The net effect is to increase the resistance to slip in the planar slip bands in high-Al alloys, hence promoting cross-slip as well as slip-on alternate planes.

Finally, the repulsion between O and Al in Ti will encourage chemical heterogeneity through segregation into Al- and O-rich subvolumes, perhaps by a spinodal mechanism. This heterogeneity should significantly perturb planar slip since the planar slip bands will traverse chemically distinct regions that have not only different resistance to dislocation glide, but different mechanisms of planar slip. This geometric heterogeneity should contribute to strain delocalization by disturbing slip bands and promoting dislocation emission from them.

Some evidence for segregation and chemical heterogeneity in Ti-6Al-0.3 O was obtained via energy-filtered TEM observations, where the diffuse scattering caused by SRO could be directly imaged by eliminating the contrast from inelastically scattered electrons[19]. To examine the degree of SRO with the presence of oxygen solutes, a Ti-6Al alloy, and Ti-6Al-0.3 O alloys were subjected to the same aging treatment (elaborated in section Materials and Methods). Figure 5 summarizes the characteristic electron diffraction patterns of different alloys. As shown in Fig. 5c, the diffraction pattern from the aged Ti-6Al alloy shows only a very faint contrast from diffuse superlattice scattering due to the lack of energy-filtering. In contrast, the aged Ti-6Al-0.3 O alloy shows strong diffuse scattering intensity despite the interference of inelastic scattering. This observation suggests a stronger ordering of aluminum with dilute oxygen addition. This

observation, in turn, suggests that there is a partitioning of oxygen and Al in the material since the domains enriched in Al would be impelled to form a higher degree of ordering to avoid Al-Al bonds. We speculate that as the reason for the higher degree of Al SRO in the aged Ti-6Al-0.3 O alloy. More recent experimental efforts reported by Gardner et al. confirms the encouragement of $\alpha_2$ formation from oxygen addition and oxygen partitioning[31–33]. Furthermore, a similar conclusion has also been made in a more comprehensive investigation about the effect of solution species (including O, Mo, and V) on $Ti_3Al$ precipitation in hcp $\alpha$-Ti by Dear et al.[34].

In conclusion, a systematic investigation of the mechanical properties was carried out to study the effect of substitutional aluminum atoms on the detrimental oxygen sensitivity of titanium. The results indicate that the oxygen embrittlement could be mitigated via the addition of aluminum, especially at cryogenic temperatures. At the microscopic level, the severe deformation localization caused by the metastable mechanical shuffling mechanism of oxygen is no longer operable with a substantial aluminum addition. The dislocation delocalization observed in the Ti-6Al-0.3 O alloys rendered a significant increase of ductility compared to Ti-0.3 O alloys while maintaining the solution strengthening of oxygen. Combined with recent theoretical and experimental reports, we proposed that the SRO of aluminum could be enhanced by oxygen, and this was confirmed with electron diffraction. With the presence of enhanced SRO, the metastable mechanical shuffling mechanism of oxygen would result in unfavorable Al-O bonds. The associated energy expense would prevent the shuffling of oxygen atoms and promote dislocation cross slip, providing extra strain-hardening ability. In addition, the deleterious {11–24} twins reported in Ti-O alloys[11,35] were also suppressed by aluminum addition. These discoveries revealed a novel strategy to mitigate the undesired sensitivity and embrittlement of interstitial species such as oxygen, which could lead to new compositions of Ti alloys with higher oxygen tolerance to reduce processing costs.

## Methods

**Materials processing and mechanical property testing**. Two Ti-O ingots with different oxygen contents (0.10 wt% and 0.30 wt%) were provided by TIMET, UK, which will be referred to as Ti-0.1 O and Ti-0.3 O hereafter. The Ti-O ingots were argon arc double melted and then forged into square bars at 1125 °C before water quenching. The bars were then hot-rolled at 800 °C and annealed at 800 °C for 1 hour. In parallel, six Ti-Al-O ingots with specifically designed combinations of aluminum contents (2.0 wt%, 4.0 wt%, and 6.0 wt%) and oxygen contents (0.10 wt% and 0.30 wt%) were also supplied by TIMET, UK. The Ti-Al-O ingots were argon arc double melted and then forged into square bars at 1125 °C before water quenching. The bars were then hot-rolled at 900 °C and annealed at 800 °C for 1 hour. Final aging treatment was conducted at 420 °C for 180 hours for the Ti-Al-O alloys to promote the SRO, if there was some. Fully recrystallized equiaxed microstructures without preferred crystallographic texture were obtained after annealing. The average grain sizes for Ti-O and Ti-Al-O alloys were ~60 μm and ~40 μm, respectively. They will be taken as the initial microstructures for subsequent mechanical property testing.

**Sample preparation and characterization method**. Dog-bone-shaped micro-tensile specimens with a gauge length of 5.0 mm, a width of 1.6 mm, and thickness of 0.8 mm were prepared from the annealed Ti-O and Ti-Al-O samples by electrical discharge machining. Uniaxial tensile tests were performed on an MTS Criterion (Model 43) system with an initial strain rate of $10^{-3} s^{-1}$ at both room temperature (~300 K) and cryogenic temperature (~77 K). For the room temperature tensile test, the digital image correlation (DIC) technique was applied to precisely measure the tensile strain, while the tensile stress was provided by the MTS system. For DIC analysis, surfaces of the micro-tensile specimens were sprayed with white and black contrast particles that served as markers for subsequent image correlation. During the tensile tests, images of the micro-tensile specimens were captured by a CCD camera with a frame rate of 100 per second. The images were later analyzed using the Vic-2D commercial software, in which the average plastic strain (Von-Mises equivalent strain) of the tensile specimen can be calculated by tracking the positions of markers. For the cryogenic temperature tensile tests, a special plastic container was designed which allowed a full

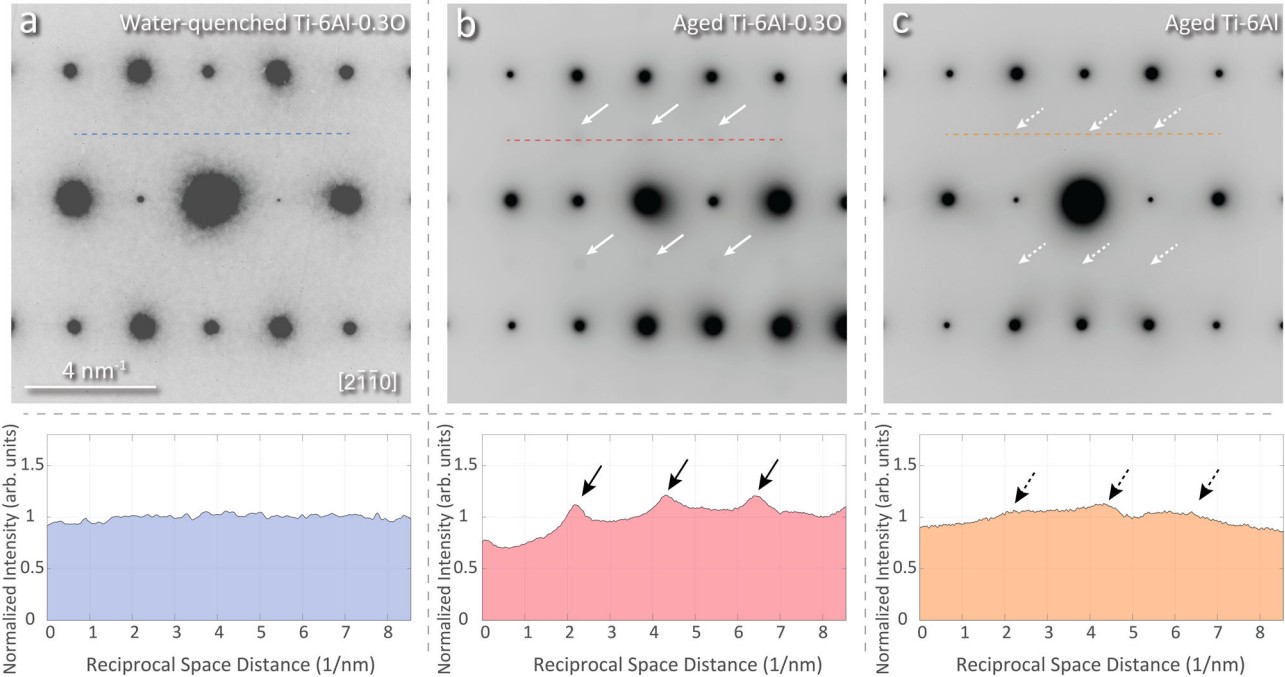

**Fig. 5 Comparison of $[2\bar{1}\bar{1}0]$ diffraction patterns for select alloys and processing conditions. a** $[2\bar{1}\bar{1}0]$ diffraction pattern of a water-quenched Ti-6Al-0.3 O alloy. **b** $[2\bar{1}\bar{1}0]$ diffraction pattern of an aged Ti-6Al-0.3 O alloy. The white arrows indicate the diffuse scattering of the Ti$_3$Al superlattice. **c** $[2\bar{1}\bar{1}0]$ diffraction pattern of an aged Ti-6Al alloy. The dashed white arrows indicate a very weak diffuse scattering of the Ti$_3$Al superlattice. The corresponding normalized intensity profiles along dotted lines in each figure are shown below.

immersion of both the micro-tensile specimens and tensile jigs in the liquid nitrogen during the tensile tests. The micro-tensile specimens were kept in liquid nitrogen for 10 mins before the onset of tensile tests. The tensile strain obtained from the MTS system was calibrated using the room temperature strain data (obtained by the DIC method), assuming that Young's modulus of the material was roughly the same at two testing temperatures. For each alloy, at least three specimens were tested at both temperatures to confirm the reproductivity of mechanical properties. To study the evolution of dislocation morphology with deformation strain, interrupted tensile experiments were also carried out at both temperatures for Ti-2Al-0.3 O, Ti-6Al-0.1 O, and Ti-6Al-0.3 O alloys. For Ti-6Al-0.1 O and Ti-6Al-0.3 O alloys, tensile deformations were interrupted at the plastic strains of 2.0%, 6.0%, and 10.0%, whereas for Ti-2Al-0.3 O alloy, tensile deformation was interrupted at the plastic strain of 2.0%.

The lateral surfaces of tensile fractured specimens were ground and mechanically polished following standard metallographic procedure. Final electro-polishing in a solution of 6% perchloric acid and 94% methanol was conducted at −40 °C using a voltage of ~30 V for 30 seconds. EBSD characterizations were carried out at areas close to the fracture surfaces, using the TSL system attached to a field emission gun scanning electron microscopy FEI Strata 235, operated at an accelerating voltage of 20 kV. The collected data were analyzed using the TSL-OIM software.

Samples for TEM investigations were prepared from the tensile fractured specimens as well as those after interrupted tensile deformations. The deformed specimens were firstly mechanically polished to a thickness of ~100 μm. Twin-jet electro-polishing was then conducted using Fischione (Model 110) at −40 °C using a voltage of ~30 V, in a solution of 6% perchloric acid and 94% methanol. TEM characterization of dislocation morphologies and electron diffraction patterns was conducted on a JEOL 3010 TEM operated at 300 kV. Weak beam dark field (WBDF) image of dislocations in Ti-6Al-0.1 O deformed by a plastic strain of 6.0 % at cryogenic temperature was carried out on the same TEM.

**Computational method**. DFT calculations are performed with Vienna ab initio Simulation Package[36], using the projector augmented wave (PAW)[37] method within the generalized gradient approximation of the exchange-correlation functional due to Perdew, Burke, and Ernzerhof (GGA-PBE)[38] including the *3p* electrons as valence for Ti. The standard PBE potential for oxygen was used for all following DFT calculations. Cutoff energy of 500 eV is used, and the Methfessel-Paxton smearing method is applied with a smearing width of 0.2 eV. An energy tolerance of $10^{-5}$ eV was used for the electron density self-consistency iterations. Ionic relaxation was terminated when forces were converged to within 0.02 eV/A for supercell relaxations; for the SFE calculations, these relaxations were terminated by an energy tolerance of $10^{-4}$ eV. Wavefunctions were sampled using a Gamma-centered k-point grid of $4 \times 3 \times 1$, where the first two grid spacings are for

directions in-plane and the last is normal to the fault plane as explained below. These settings guarantee total energy convergence within 1 meV/atom.

The bulk supercell is constructed with lattice vectors parallel to $x = [2-1-10]$, $y = [01-10]$, and $c = [0001]$, see Fig. S3a for reference. The dimension along x, y, and z are three, two, and five times the orthogonal bulk hcp cell, respectively. SQSs[39] of Ti-10at%Al with 120 atoms are generated by the mcsqs code within Alloy Theoretic Automated Toolkit package[40]. The cell is fully relaxed with respect to atomic position, cell volume, and shape. To impose an I$_2$ stacking fault, each of the 10 possible planes is picked and rigidly shifted along a 1/3 <10-10> lattice direction by imposing a tilt in the cell to generate the fault. The atomic positions within the supercell are fully relaxed while the cell volume and shape are held fixed. The SFE at each fault plane is calculated by subtracting the energies of the tilted cell and the undeformed cell divided by the slip plane area per supercell. Similar steps followed for the pure Ti cell with the same supercell dimension to compute the I$_2$ SFE without any solutes.

Calculations of the interaction energy between a substitutional aluminum solute and an interstitial oxygen atom were performed using 360-atom Ti atom supercell. A single Al atom is used in supercells with dimensions of $6 \times 6 \times 5$ along the lattice directions of an HCP unit cell with a $3 \times 3 \times 1$ k-point mesh. The interaction energies between the aluminum and a single oxygen atom in both octahedral and hexahedral interstitial positions at different neighboring distances are calculated. The difference in site energies is defined as the difference between the energy of the supercell with an Al substitutional atom and oxygen at an interstitial site and that of reference, the cell with oxygen at the octahedral position and an aluminum atom located furthest from each other. The convergence criterion and the rest of the DFT settings for these calculations are the same as used for the SFE calculations.

## Data availability

The mechanical data generated in this study have been deposited in the Figshare database under accession code: https://figshare.com/articles/dataset/Pure_Ti_tensile_data_at_RT_and_LN2/16627903.

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

## Acknowledgements

We gratefully acknowledge funding from the US Office of Naval Research under grant no. N00014-19-1-2376. Work at the Molecular Foundry was supported by the Office of Science, Office of Basic Energy Sciences, of the US Department of Energy under Contract no. DE-AC02-05CH11231. We would like to thank TIMET for the production of the model alloys. Computational resources were provided by the Extreme Science and Engineering Discovery Environment (XSEDE), which is supported by National Science Foundation grant number ACI-1548562 and the Savio computational cluster resource provided by the Berkeley Research Computing program at the University of California, Berkeley (supported by the UC Berkeley Chancellor, Vice Chancellor for Research, and Chief Information Officer).

## Author contributions

Y.C., R.Z., J.W.M., and A.M.M. prepared the manuscript, which was reviewed and edited by all authors; Y.C., R.Z., and S.Z. conducted the tensile tests; Y.C. and R.Z. analyzed the tensile data; Y.C. and R.Z. conducted TEM experiments; Y.C. conducted EBSD experiments; M.H. D.C.C. and M.A. conducted the calculations of stacking fault energy. Project administration, supervision, and funding acquisition were performed by M.A., D.C.C., J.W.M., and A.M.M.

## Competing interests

The authors declare no competing interests.
