## [Peer Review File · Nature Communications]

Title: Elimination of oxygen sensitivity in α -titanium by substitutional alloying with AlREVIEWER COMMENTS

Reviewer #1 (Remarks to the Author):

Comments to the authors:

This is in principle a good paper, part of a series which these authors have been publishing and is generally of high quality. However, there is a number of specific claims that are central to their mechanistic conclusions that are not substantiated. Thus, from the abstract and body of the manuscript, the authors claim the mechanism involves the following:

i. The change in dislocation pattern is due to cross-slip of dislocations. The authors have not actually shown that cross-slip has occurred. A simple experiment would involve crystallographic analysis showing a given dislocation (of known Burgers vector determined by diffraction contrast) lying on a primary plane (the plane determined by the cross-product of the Burgers vector and the line direction (determined by crystallographic analysis)), and then lying on the cross-slip plane (also identified by the cross-product of the new line direction and the Burgers vector).

ii. There is simultaneous glide of dislocations on basal and prismatic planes. The authors do not show images of dislocations gliding on the basal and prismatic planes. This will require unique Burgers determination (full diffraction contrast, not just one image recorded with $g=0002$) and line directions (determined using crystallographic analysis), so that the subsequent cross-products would establish unambiguously that both slip planes are active.

iii. Reference is made to nodes of tangled dislocations acting as Frank-Read sources emitting loops. The statement is made as if there is direct evidence of this. This is not evident from the images presented. If the evidence is available, it should be presented with detailed diffraction contrast and crystallographic analyses. If the evidence is "consistent with", then this should be so stated.

Other comments:

i. The authors refer to individual O interstitials lying on a slip plane as being obstacles to dislocation motion. Is this meant to be different from solid solution strengthening?

ii. The claim that this work points the way for low-cost Ti alloys is somewhat over the top. The applications of Ti alloys is mainly at ambient and intermediate temperatures, not under cryogenic conditions. I am not able to believe that the next generation of Ti alloys will all contain 6Al and 0.3O! I believe that this speculation should be removed from the paper – the content is already of very high quality, and this unsubstantiated claim detracts from this quality.

iii. Change the term, "g dot b" analysis to "diffraction contrast".

Summary: A good paper that should be published after the points mentioned above have been dealt with.

Reviewer #2 (Remarks to the Author):

In this article, the authors find new and valuable results, for example, 6-fold increase of ductility in Ti-0.3O alloys by the dislocation delocalization in adding 6% Al, and a suppression of the deleterious {11-24} twins in Ti-O alloys by aluminum addition.

However, I think the discussions on the deformation mechanism of Ti-Al-O alloys is not sufficient at this moment. The effects of Al and O solutes on the deformation behavior of Ti-Al-O alloys are discussed using EBSD-IPF & IQ maps shown in Fig. 2. The authors described the deformation twinning formation in detail when Al and O contents are changed, but they are not enough to lead the conclusions. This is because the inside of α -Ti grains containing both Al and O solutes of each sample are focused, but no investigation on the roles of the grain boundaries and orientations on the plastic deformation behaviors. Therefore, I recommend them to show a kernel average misorientation (KAM) value, showing local grain misorientation, of each sample after tensile testing in liquid nitrogen to analyze the induced strains at grain boundaries and discuss their effects on the deformation behavior of Ti-Al-O alloys with different Al and O contents.

I want to review the additional results and discussions on the above points again.

Reviewer #3 (Remarks to the Author):

The manuscript presents surprising results on a coupling effect between Al and O solutes in Ti at cryogenic temperatures whereby the addition of both elements improves the ductility and tensile strength over the inclusion of only one which tends to limit ductility. The authors propose several mechanisms which lead to this rare increase in both ductility and strength which are supported by EBSD, SEM, TEM, DFT, and electron diffraction. As is noted in the supplemental material, this enhancement of ductility does not seem to hold at room temperature for reasons the authors discuss.

The manuscript is well written, the results interesting, and the analysis sound. Sufficient detail is provided to reproduce the results presented. The work does raise a few questions however, some of which are clearly beyond the scope of the current work, but a few which might be clarified in the current paper.

What is the limit at which this increased ductility and strength might be reached at LN2 temperatures? The concentration of aluminum could probably be doubled or more with only the SRO being effected in terms of the microstructure and oxygen has an incredibly high solubility in Ti. This is beyond the scope of the current paper to investigate, but a comment on what might limit this behavior given the depth with which the mechanisms have been considered would be interesting.

Second, the authors mention two different mechanisms related to the reported repulsion between Al and O in Ti lattices: first that the shuffle of oxygen to a hexahedral site should be impeded if the hexahedral site has Al neighbors (and the recovery to the octahedral site made easier), second that the repulsion should create heterogeneous regions of higher Al and O content respectively. To the first possibility, this could be easily supported by a straight-forward DFT calculation of the change in hexahedral interstitial energy when Al is a nearest neighbor. The results from reference [21] suggest this possibility, but that paper considered only octahedral interstitials. To the second possibility, the same reference indicates that an order Ti_6Al_2O phase, where O is present as an interstitial within the ordered Ti_3Al phase should be a ground state. If this is a stable ground state, it is unclear how the presence of oxygen could promote segregation in the way suggested. It may also be worth considering the plastic behavior of this phase.

As mentioned above, some of these issues are beyond the scope of the paper. Again, the paper, its methods, and its conclusions are sound and interesting. I can recommend it for publication.

Reviewer #1 (Remarks to the Author):

Comments to the authors:

This is in principle a good paper, part of a series which these authors have been publishing and is generally of high quality. However, there is a number of specific claims that are central to their mechanistic conclusions that are not substantiated. Thus, from the abstract and body of the manuscript, the authors claim the mechanism involves the following:

We appreciate the overall constructive and detailed feedback from the reviewer. Our responses and revisions are presented below.

i. The change in dislocation pattern is due to cross-slip of dislocations. The authors have not actually shown that cross-slip has occurred. A simple experiment would involve crystallographic analysis showing a given dislocation (of known Burger's vector determined by diffraction contrast) lying on a primary plane (the plane determined by the cross-product of the Burgers vector and the line direction (determined by crystallographic analysis)), and then lying on the cross-slip plane (also identified by the cross-product of the new line direction and the Burgers vector).

Response to reviewer's comment i: We agree with the reviewer that a systematic crystallographic analysis is necessary to accurately characterize an individual cross-slip event. However, there are some practical limitations for doing this type of analysis in our samples.

1. Most observed dislocations in the alloys featured in the current manuscript are screw- or near screw-oriented, which makes it not possible to apply the cross-product method to determine the slip plane.
2. What evidence we do believe is indicative of cross slip is shown in Fig. 3. Here, the dislocation interactions and entanglements that exist outside of the primary planar slip bands were formed at relatively late stage of the deformation. Therefore, at this late stage the high dislocation density obstructs the acquisition of a tilting series that could potentially be used to determine the slip planes of the entangled dislocations. We made some trials of weak-beam imaging for **dislocation tomography**, but the complex dislocation entanglements are too concentrated for a reliable reconstruction.

The primary slip plane of the planar slip bands in the alloys can be easily determined by doing trace analysis of the slip band. As shown in Fig. 4, due to the low symmetry of hcp systems, we can determine the Burgers vectors and the primary slip planes of the planar slip bands with the g - $3g$ weak-beam images taken with $g = (0002)$ and $g = (01-10)$ near the $[2-1-10]$ orientation. At these conditions, basal plane is perpendicular to the imaging plane, therefore the observed planar slip bands consist of prismatic slip. According to the invisibility condition, the possible Burgers vectors are $1/3[11-20]$ and $1/3[1-210]$. So, the dislocations in the slip bands shown in Fig. 4 are $1/3[1-210]$ a-type dislocations gliding on $(10-10)$ prismatic planes or $1/3[11-20]$ a-type dislocations gliding on $(1-100)$ prismatic planes. A crystal schematic was also added to Fig. 4 to better illustrate the observation direction.

Our statement related to the assertion of cross-slip events is based on the observed jagged dislocations and the extensive dislocation entanglement. As the reviewer pointed out, this is indirect evidence, but it would be difficult to rationalize the jaggedness of these a-type dislocations in another way. We modified the manuscript to reflect the reasoning.

ii. There is simultaneous glide of dislocations on basal and prismatic planes. The authors do not show images of dislocations gliding on the basal and prismatic planes. This will require unique Burgers determination (full diffraction contrast, not just one image recorded with $g=0002$) and line directions (determined using crystallographic analysis), so that the subsequent cross-products would establish unambiguously that both slip planes are active.

Response to reviewer's comment ii: As we detailed in the previous response comment, the primary slip plane of the planar slip bands in the alloys can be identified by doing trace analysis of the slip band. As shown in Fig. 4, due to the low symmetry of hcp systems, we can determine the Burgers vectors and the primary slip planes of the planar slip bands with the g - $3g$ weak-beam images taken with $g = (0002)$ and $g = (01-10)$ near the $[2-1-10]$ orientation. At this condition, the basal plane is perpendicular to the imaging plane, therefore the observed planar slip bands consist of prismatic slip. According to the invisibility condition, the possible Burgers vectors are $1/3[11-20]$ and $1/3[1-210]$. So, the dislocations in the slip bands shown in Fig. 4 are $1/3[1-210]$ a-type dislocations gliding on $(10-10)$ prismatic planes or $1/3[11-20]$ a-type dislocations gliding on $(1-100)$ prismatic planes.

Unfortunately, we did not locate suitable basal planar slip bands for imaging.

iii. Reference is made to nodes of tangled dislocations acting as Frank-Read sources emitting loops. The statement is made as if there is direct evidence of this. This is not evident from the images presented. If the evidence is available, it should be presented with detailed diffraction contrast and crystallographic analyses. If the evidence is "consistent with", then this should be so stated.

Response to reviewer's comment iii: We appreciate the reviewer's comment on this discussion. It is true that we do not have direct evidence of Frank-Read sources generating dislocations. We have modified the manuscript in page 10 as follows,

More importantly, the nodes of dislocation tangles are consistent with the theory that they can act as Frank-Read sources to emit dislocation loops into the undeformed areas in-between these slip bands.

Other comments:

iv. The authors refer to individual O interstitials lying on a slip plane as being obstacles to dislocation motion. Is this meant to be different from solid solution strengthening?

Response to reviewer's comment iv: We appreciate the reviewer's comment on this topic. This is indeed a critical question related to the oxygen sensitivity of titanium. Solid solution strengthening usually refers to the interaction of the strain fields of dislocations with the strain fields with the solute atoms. But the interactions we referred to in the manuscript is beyond solid solution strengthening, where the slip of dislocations will mechanically shuffle the crystallographic sites of oxygen atoms. Thus, this effect is different than conventional solid solution strengthening. We modified the manuscript in page 2 to reflect this point.

Compared to a strain-field interaction via solid solution strengthening, the shuffling mechanism will mechanically shuffle the crystallographic sites of oxygen atoms, leading to strong oxygen sensitivity as described in our prior work [11].

v. The claim that this work points the way for low-cost Ti alloys is somewhat over the top. The applications of Ti alloys is mainly at ambient and intermediate temperatures, not under cryogenic conditions. I am not able to believe that the next generation of Ti alloys will all contain 6Al and 0.3O! I believe that this speculation should be removed from the paper – the content is already of very high quality, and this unsubstantiated claim detracts from this quality.

Response to reviewer's comment v: We agree with the reviewer, this is a fair comment. The current manuscript is primarily focused on the deformation mechanism of the featured alloys. Any engineering applications would need further investigations and consideration of other factors such as cost. Accordingly, we have modified the final sentence of the conclusion as follows:

"These discoveries revealed a novel strategy to mitigate the undesired sensitivity and embrittlement of interstitial species such as oxygen, which could lead to new compositions of Ti alloys with higher oxygen tolerance to reduce processing costs."

vi. Change the term, "g dot b" analysis to "diffraction contrast".

Response to reviewer's comment vi: We appreciate the reviewer's comment and have modified the terminology accordingly.

Summary: A good paper that should be published after the points mentioned above have been dealt with.

We appreciate the reviewer's comments that strengthened our manuscript.

Reviewer #2 (Remarks to the Author):

In this article, the authors find new and valuable results, for example, 6-fold increase of ductility in Ti-0.3O alloys by the dislocation delocalization in adding 6% Al, and a suppression of the deleterious {11-24} twins in Ti-O alloys by aluminum addition.

However, I think the discussions on the deformation mechanism of Ti-Al-O alloys is not sufficient at this moment. The effects of Al and O solutes on the deformation behavior of Ti-Al-O alloys are discussed using EBSD-IPF & IQ maps shown in Fig. 2. The authors described the deformation twinning formation in detail when Al and O contents are changed, but they are not enough to lead the conclusions. This is because the inside of α -Ti grains containing both Al and O solutes of each sample are focused, but no investigation on the roles of the grain boundaries and orientations on the plastic deformation behaviors.

Therefore, I recommend them to show a kernel average misorientation (KAM) value, showing local grain misorientation, of each sample after tensile testing in liquid nitrogen to analyze the induced strains at grain boundaries and discuss their effects on the deformation behavior of Ti-Al-O alloys with different Al and O contents.

I want to review the additional results and discussions on the above points again.

Response to reviewer's comment: We truly appreciate the reviewer's comment on the importance of evaluating the roles of grain boundaries in plastic deformation of Ti-Al-O alloys through EBSD-KAM maps.

For this purpose, we characterized the areas near the fracture surfaces (77 K) of Ti-Al-O samples. The step size of EBSD scan was 0.05 μm , and KAM values were calculated considering the first nearest neighbor. The obtained inverse pole figure (IPF) maps and corresponding KAM maps are shown in the figure below (Fig. R1), in which the fractured strain was indicated for each microstructure. The KAM map range was 0~5°.

In the Ti-2Al-0.1O sample, a large number of deformation twins was observed, and the KAM values at the grain boundaries and twin boundaries were generally higher than those inside the grains. This reflected a dominant contribution of deformation twins to the excellent strain-hardening ability of Ti-2Al-0.1O at 77 K. With further increase of Al content in low oxygen Ti-Al-O alloys, the deformation twins were gradually suppressed. The plastic deformation was more localized at the grain boundaries, as reflected by the generally larger KAM values at the grain boundaries in Ti-4Al-0.1O and Ti-6Al-0.1O alloys.

In high oxygen Ti-Al-O alloys, the plastic deformation at grain boundaries remained comparably larger than that inside the grains. However, one clear tendency is that, with increasing Al content, the plastic deformation (reflected by the KAM values) became progressively larger inside the grains. This is particularly evident when comparing the KAM maps of Ti-2Al-0.3O and Ti-6Al-0.3O. The distribution of KAM values clearly became more homogenous in the latter.

In the manuscript, we specifically chose Ti-2Al-0.3O, Ti-6Al-0.1O and Ti-6Al-0.3O alloys in which the deformation twins were all basically suppressed, and discussed the contribution of dislocation cross-slip to the strain-hardening abilities of the three alloys. We concluded that the disruption of planar slip and increasing higher propensity of dislocation cross-slip from Ti-2Al-0.3O, Ti-6Al-0.1O to Ti-6Al-0.3O was the main reason for the gradually increased strain-hardening rate among the three alloys. This can also be supported by KAM map characterizations in Fig. R1. It is clearly shown that the distribution of KAM values (indicator of plastic deformation) became more and more homogenous, from Ti-2Al-0.3O, Ti-6Al-0.1O to Ti-6Al-0.3O alloys. This can be beneficial for the ductility in two ways. Firstly, the substantial dislocation interactions/cross-slips can provide sufficient strain-hardening ability to sustain the high stress level at 77 K. Secondly, the strain localization at the grain boundaries can be somehow relaxed, preventing/delaying the formation of grain boundary micro-cracks.

We have added these discussions as supplementary text and Fig. R1 as a new supplementary Figure S2 in the revised manuscript. Some of the discussions are also added to the main text of the revised manuscript on Page 6.

Fig. R1 IPF maps and corresponding KAM maps of Ti-Al-O alloys tensile fractured at 77 K. The range of KAM value is 0~5°. The fracture strain is indicated for each microstructure. The tensile direction is horizontal for all microstructures.

Reviewer #3 (Remarks to the Author):

The manuscript presents surprising results on a coupling effect between Al and O solutes in Ti at cryogenic temperatures whereby the addition of both elements improves the ductility and tensile strength over the inclusion of only one which tends to limit ductility. The authors propose several mechanisms which lead to this rare increase in both ductility and strength which are supported by EBSD, SEM, TEM, DFT, and electron diffraction. As is noted in the supplemental material, this enhancement of ductility does not seem to hold at room temperature for reasons the authors discuss.

The manuscript is well written, the results interesting, and the analysis sound. Sufficient detail is provided to reproduce the results presented. The work does raise a few questions however, some of which are clearly beyond the scope of the current work, but a few which might be clarified in the current paper.

We truly appreciate the reviewer's positive feedback on our current study and constructive suggestions for our future studies. The detailed responses and considerations addressing the reviewer's comments are shown as follows:

1. What is the limit at which this increased ductility and strength might be reached at LN₂ temperatures? The concentration of aluminum could probably be doubled or more with only the SRO being affected in terms of the microstructure and oxygen has an incredibly high solubility in Ti. This is beyond the scope of the current paper to investigate, but a comment on what might limit this behavior given the depth with which the mechanisms have been considered would be interesting.

Response to reviewer's comment: We agree with the reviewer that it is interesting to consider the limit of increased strength and ductility of Ti-Al-O alloys at LN₂ temperature, which is instructive for designing our future work. Indeed, we are considering to further increase the oxygen content (up to ~0.5 wt.%) and aluminum content (up to ~10 wt.%), for probing the upper limit of interstitial/substitutional alloying in α -titanium without a significant loss of ductility.

Concerning the strength, it is reasonable to predict that it will continuously increase with increasing oxygen and aluminum contents, due to the solid solution hardening effect, as well as Ti₃Al (α_2) precipitates at aluminum content higher than 6 wt.%. Ti-O precipitation might also have a similar effect.

Concerning the ductility, however, it is difficult to predict based on two possible reasons. Firstly, at aluminum content higher than 6 wt.%, SRO becomes less dominant in the microstructure, which is expected to be replaced by α_2 precipitates. The effects of fine (or even nano-sized) α_2 precipitates on the tensile ductility is not clear. Secondly, we are conservative on whether an even higher oxygen content (i.e., ~0.5 wt.%) will further increase the ductility of Ti-Al-O alloys at 77 K. Since the oxygen has a rather potent hardening effect, which means that an even higher strain-hardening rate is necessary to maintain the same uniform elongation, according to the Considere's plastic instability Criterion ($d\sigma/d\varepsilon \leq \sigma$, in which σ and ε are true stress and true strain, respectively, $d\sigma/d\varepsilon$ is the strain-hardening rate). The strain-

hardening ability of Ti-Al-O alloy generally comes from the dislocation activities, i.e. dislocation cross-slip. This means an even higher dislocation density and more propensity of dislocation cross-slip are necessary to maintain the same uniform elongation/ductility in the higher oxygen sample. Another disadvantage of higher oxygen content could be related to the possibility of grain boundary embrittlement. Although the addition of Al has substantially suppressed the grain boundary embrittlement in Ti-6Al-0.3O alloy, we might see the reemergence of this limiting/detrimental behavior at a higher oxygen content. In addition, at high oxygen content, the precipitation of Ti_6O and/or Ti_6Al_2O phases are expected at least at lower temperatures. It is not clear whether these would be beneficial to the ductility.

Above all, we appreciate the reviewer's instructive comment, and we will try to find out the upper limit of good strength/ductility synergy in Ti-Al-O system at 77 K in our future work. However, we don't believe that we know enough about this topic to speculate in this current manuscript.

2. Second, the authors mention two different mechanisms related to the reported repulsion between Al and O in Ti lattices: first that the shuffle of oxygen to a hexahedral site should be impeded if the hexahedral site has Al neighbors (and the recovery to the octahedral site made easier), second that the repulsion should create heterogeneous regions of higher Al and O content respectively. To the first possibility, this could be easily supported by a straight-forward DFT calculation of the change in hexahedral interstitial energy when Al is a nearest neighbor. The results from reference [21] suggest this possibility, but that paper considered only octahedral interstitials.

Response to reviewer's comment: In response to this comment, we performed additional DFT calculations of the interaction energy between an aluminum substitutional solute and interstitial oxygen in both octahedral and hexahedral positions (Fig. R2 below). For the octahedral interstitials the results are largely consistent with those in Ref. [21]. Specifically, our results indicate that, relative to the lowest energy position of the oxygen atom, in an octahedral position far from the Al solute, there is a repulsive interaction energy for the nearest neighbor site of 0.71 eV, which is reduced to 0.03 eV at the second neighbor octahedral site (these results agree qualitatively with those in Ref. [21] although the interaction energies are 0.05 eV lower presumably due to differences in the supercell geometries). By comparison, the interaction of Al and oxygen in the nearest neighbor hexahedral site shows the highest repulsive value of all sites considered, with a magnitude of 1.60 eV. The interaction energy for the second neighbor hexahedral site is 1.46 eV, while the energy difference between octahedral and hexahedral is 1.10 eV in bulk Ti from the Al solute. We interpret these results to have the following consequences for the experimental results obtained in the present work. Prior to deformation, we expect that oxygen atoms would be positioned at second neighbor octahedral sites to Al or further, due to the large positive interaction energy that is much higher than kT for relevant heat treatment temperatures. Thus, if a shuffle of oxygen brings it to a nearest neighbor hexahedral position the energy of the system would be raised by 1.57 eV, compared to a value of 1.10 eV in pure Ti. This should impede the shuffle to a nearest-neighbor site of Al (and make recovery back to the second-neighbor site easier) relative to the situation for bulk Ti. We note also that if oxygen resides in a short-range-ordered domain with a structure similar to the α_2 - Ti_3Al ordered phase, oxygen would be expected to sit in the octahedral sites surrounded by Ti, and shuffle from these sites to any hexahedral neighboring site would result in oxygen residing in a site with a nearest-

neighbor Al – based on the results summarized above this scenario would be expected to lead to an increase in energy larger than in bulk Ti.

To the second possibility, the same reference indicates that an order Ti_6Al_2O phase, where O is present as an interstitial within the ordered Ti_3Al phase should be a ground state. If this is a stable ground state, it is unclear how the presence of oxygen could promote segregation in the way suggested. It may also be worth considering the plastic behavior of this phase.

Response to reviewer's comment: We agree with the referee that with high enough oxygen content, formation of the Ti_6Al_2O phase would be expected, provided enough oxygen was present for the alloy to reside on a tie-line involving this phase, or inside a tie triangle involving this phase. However, in the present systems studied, there is no evidence for precipitation and the alloys reside in a single-phase solid solution state. It becomes desirable then to speculate about the nature of the SRO and whether this would be in the form of ordered Ti_6Al_2O like domains (with preference for Al-Al second neighbors and Ti-O nearest-neighbors) or whether instead the alloy would prefer to form Ti_3Al ordered domains and separate Ti_6O like ordered domains. The latter case would involve segregation like that mentioned in the manuscript. The answer to this question would be determined by (i) the thermodynamic driving forces for SRO formation, and (ii) the kinetics of the oxygen and Al diffusion. Concerning (i) we note that the constant-oxygen-chemical-potential phase diagrams in Ref. [21] show that for systems inside the two-phase regions between α and α_2 , oxygen segregates not to the α_2 phase, but rather to the solid solution (α). Additionally, at very low temperature the results from Ref. [21] suggest that our alloys have a composition that lies within a three-phase triangle between α and α_2 and Ti_6O , such that segregation of oxygen away from the Al-rich intermetallic is again expected. Based on these considerations, it would seem that thermodynamics would favor segregation of oxygen out of the α_2 ordered domains into the solid solution, consistent with what is proposed in the manuscript. By contrast, such segregation may require longer-ranged diffusion of oxygen relative to Al and could be disfavored relative to the formation of Ti_6Al_2O like ordered domains that would presumably not require such interdiffusion. This latter scenario could in principle occur if rapid quenches were made from a high-temperature random state. However, the alloys were shown to form significant SRO, such that kinetics was apparently sufficient to allow for significant diffusion of the slower-diffusing Al substitutional atoms. Detailed kinetic-Monte-Carlo simulations or atom-probe-microscopy experiments would be needed to provide additional insight, but we suspect that we may be in a regime where thermodynamics would favor the segregation of oxygen away from the ordered α_2 -like domains.

In response to these comments, we have added a new figure (Fig. S4) to the Supplemental Material, presenting the results of the additional DFT calculations for the interaction energy between Al and O interstitials in both octahedral and hexahedral positions. In addition, we have modified the discussion section in page 14 to read:

The shuffle will be impeded if the hexahedral site has Al neighbors, hence increasing the resistance to dislocation glide. Moreover, the higher energy of the hexahedral site with Al neighbors will decrease the activation energy for the return of the interstitial to its original octahedral site, removing the impetus for planar slip. This picture is supported by DFT calculations presented in Fig. S4, as discussed in Supplementary text. The net effect is to increase the resistance to slip in the planar slip bands in high-Al alloys, hence promoting cross-slip as well as slip on alternate planes.

Fig. R2: DFT calculations of the interaction energy between an isolated Al solute atom and oxygen at different neighboring positions in both the octahedral site (black circles) and hexahedral site (blue squares).

As mentioned above, some of these issues are beyond the scope of the paper. Again, the paper, its methods, and its conclusions are sound and interesting. I can recommend it for publication.

REVIEWERS' COMMENTS

Reviewer #1 (Remarks to the Author):

Comments on the authors responses to Reviewer #1

1. Response to my first comment:

We agree with the reviewer that a systematic crystallographic analysis is necessary to accurately characterize an individual cross-slip event. However, there are some practical limitations for doing this type of analysis in our samples.

1. Most observed dislocations in the alloys featured in the current manuscript are screw- or near screw-oriented, which makes it not possible to apply the cross-product method to determine the slip plane.
2. What evidence we do believe is indicative of cross slip is shown in Fig. 3. Here, the dislocation interactions and entanglements that exist outside of the primary planar slip bands were formed at relatively late stage of the deformation. Therefore, at this late stage the high dislocation density obstructs the acquisition of a tilting series that could potentially be used to determine the slip planes of the entangled dislocations. We made some trials of weak-beam imaging for dislocation tomography, but the complex dislocation entanglements are too concentrated for a reliable reconstruction.

My new response: I understand the difficulty with analyzing the cross-slip segments if they are mainly screws, and if they are obscured by the high dislocation density. What about the long segments in Fig. 4(d) – which are clearly visible and cannot all be screw segments?

The primary slip plane of the planar slip bands in the alloys can be easily determined by doing trace analysis of the slip band. As shown in Fig. 4, due to the low symmetry of hcp systems, we can determine the Burgers vectors and the primary slip planes of the planar slip bands with the g -3 g weak-beam images taken with $g = (0002)$ and $g = (01-10)$ near the $[2-1-10]$ orientation. At these conditions, basal plane is perpendicular to the imaging plane, therefore the observed planar slip bands consist of prismatic slip. According to the invisibility condition, the possible Burgers vectors are $1/3[11-20]$ and $1/3[1-210]$. So, the dislocations in the slip bands shown in Fig. 4 are $1/3[1-210]$ a-type dislocations gliding on $(10-10)$ prismatic planes or $1/3[11-20]$ a-type dislocations gliding on $(1-100)$ prismatic planes. A crystal schematic was also added to Fig. 4 to better illustrate the observation direction.

My new response: I have difficulty with the crystallography – in Fig. 4(c), the $[01-10]$ direction (from the g -vector) is nearly parallel to the slip bands. If the slip planes are as claimed, should that angle be closer to 30° ? Just a question of interest, I am sure that the authors have thought about this in more detail than have I!

Our statement related to the assertion of cross-slip events is based on the observed jagged dislocations and the extensive dislocation entanglement. As the reviewer pointed out, this is indirect evidence, but it would be difficult to rationalize the jaggedness of these a-type dislocations in another way. We modified the manuscript to reflect the reasoning.

My new response: Thanks.

Response to my second comment:

Response to reviewer's comment ii: As we detailed in the previous response comment, the primary slip plane of the planar slip bands in the alloys can be identified by doing trace analysis of the slip band. As shown in Fig. 4, due to the low symmetry of hcp systems, we can determine the Burgers vectors and the primary slip planes of the planar slip bands with the g -3 g weak-beam images taken with $g = (0002)$ and $g = (01-10)$ near the $[2-1-10]$ orientation. At this condition, the basal plane is perpendicular to the imaging plane, therefore the observed planar slip bands consist of prismatic slip. According to the invisibility condition, the possible Burgers vectors are $1/3[11-20]$ and $1/3[1-210]$. So, the dislocations in the slip bands shown in Fig. 4 are $1/3[1-210]$ a-type dislocations gliding on $(10-10)$ prismatic planes or $1/3[11-20]$ a-type dislocations gliding on $(1-100)$ prismatic planes.

Unfortunately, we did not locate suitable basal planar slip bands for imaging.

My new response: Are you still claiming prismatic and basal slip?

Response to my third comment:

Response to reviewer's comment iii: We appreciate the reviewer's comment on this discussion. It is true that we do not have direct evidence of Frank-Read sources generating dislocations. We have modified the manuscript in page 10 as follows,

More importantly, the nodes of dislocation tangles are consistent with the theory that they can act as Frank-Read sources to emit dislocation loops into the undeformed areas in-between these slip bands

My new response: Good response.

Response to other comments:

iv. The authors refer to individual O interstitials lying on a slip plane as being obstacles to dislocation motion. Is this meant to be different from solid solution strengthening?

Response to reviewer's comment iv: We appreciate the reviewer's comment on this topic. This is indeed a critical question related to the oxygen sensitivity of titanium. Solid solution strengthening usually refers to the interaction of the strain fields of dislocations with the strain fields with the solute atoms. But the interactions we referred to in the manuscript is beyond solid solution strengthening, where the slip of dislocations will mechanically shuffle the crystallographic sites of oxygen atoms. Thus, this effect is different than conventional solid solution strengthening. We modified the manuscript in page 2 to reflect this point.

My new response: Solid solution strengthening is a result of not only interaction with strain fields, but also local elastic modulus changes, and the possibility of promotion of some directional bonding which can result in strengthening. Any consideration of these effects?

v. The claim that this work points the way for low-cost Ti alloys is somewhat over the top. The

applications of Ti alloys is mainly at ambient and intermediate temperatures, not under cryogenic conditions. I am not able to believe that the next generation of Ti alloys will all contain 6Al and 0.3O! I believe that this speculation should be removed from the paper – the content is already of very high quality, and this unsubstantiated claim detracts from this quality.

Response to reviewer's comment v: We agree with the reviewer, this is a fair comment. The current manuscript is primarily focused on the deformation mechanism of the featured alloys. Any engineering applications would need further investigations and consideration of other factors such as cost.

Accordingly, we have modified the final sentence of the conclusion as follows:

“These discoveries revealed a novel strategy to mitigate the undesired sensitivity and embrittlement of interstitial species such as oxygen, which could lead to new compositions of Ti alloys with higher oxygen tolerance to reduce processing costs.”

My new response: Thanks.

vi. Change the term, "g dot b" analysis to "diffraction contrast".

Response to reviewer's comment vi: We appreciate the reviewer's comment and have modified the terminology accordingly.

My new response: Thanks.

Summary: I remain with my opinion that this is a good paper that should be published. I have raised questions in my responses to their responses, and I think it would be advantageous to the authors if they would take these comments into consideration. However, I do not need to see their responses before recommending acceptance of the paper.

Reviewer #2 (Remarks to the Author):

The authors completely showed the suitable answers and detail explanation to my questions and comments on this article.

I think the revised manuscript should be published by this journal.

Reviewer #3 (Remarks to the Author):

The authors' have done a thorough job in responding to the previous reviewer comments. It can now be recommended for publication.

Comments on the authors responses to Reviewer #1

Blue text = our previous response

Black text = Review 1 new comments

Red text= our response to the new comments of the reviewer

1. Response to my first comment:

We agree with the reviewer that a systematic crystallographic analysis is necessary to accurately characterize an individual cross-slip event. However, there are some practical limitations for doing this type of analysis in our samples.

1. Most observed dislocations in the alloys featured in the current manuscript are screw- or near screw-oriented, which makes it not possible to apply the cross-product method to determine the slip plane.
2. What evidence we do believe is indicative of cross slip is shown in Fig. 3. Here, the dislocation interactions and entanglements that exist outside of the primary planar slip bands were formed at relatively late stage of the deformation. Therefore, at this late stage the high dislocation density obstructs the acquisition of a tilting series that could potentially be used to determine the slip planes of the entangled dislocations. We made some trials of weak-beam imaging for **dislocation tomography**, but the complex dislocation entanglements are too concentrated for a reliable reconstruction.

Reviewer 1 new response: I understand the difficulty with analyzing the cross-slip segments if they are mainly screws, and if they are obscured by the high dislocation density. What about the long segments in Fig. 4(d) – which are clearly visible, and cannot all be screw segments?

Response to the reviewer's new response: We appreciate the reviewer's understanding of the experimental difficulties as well as the suggested experiments. However, there are 3 issues that preclude us from making any further determinations about the nature of the dislocations between the slip bands in Figure 4:

1) The dislocation density, even for the regions between the different planar slip bands, are only relatively lower than in the bands. We can distinguish individual dislocations in Fig. 4 (d) as it was specifically chosen to give us the best g-3g contrast. One can refer to Fig. 3 (e) for the more general dislocation contrast in a 2-beam condition.

2) Even for the region shown in Fig. 4 (d), the entangled dislocations are causing difficulties as the speculated cross slipped sections are mostly in the heavily curved and entangled sections.

3) The situation is even worse for the Ti-6Al-0.3O samples, where the delocalization appears even earlier.

In light of these three experimental issues, we believe a full analysis of the dislocations outside of the slip bands in this condition are beyond the scope of this manuscript. We do plan to conduct more systematic studies of the cross-slip activities in the alloys with lower amounts of deformation in the future projects. The caption of Fig. 3 and Fig 4 are updated to reflect the aforementioned experimental limitations.

The primary slip plane of the planar slip bands in the alloys can be easily determined by doing trace analysis of the slip band. As shown in Fig. 4, due to the low symmetry of hcp systems, we can determine

the Burgers vectors and the primary slip planes of the planar slip bands with the g - $3g$ weak-beam images taken with $g = (0002)$ and $g = (01-10)$ near the $[2-1-10]$ orientation. At these conditions, basal plane is perpendicular to the imaging plane, therefore the observed planar slip bands consist of prismatic slip. According to the invisibility condition, the possible Burgers vectors are $1/3[11-20]$ and $1/3[1-210]$. So, the dislocations in the slip bands shown in Fig. 4 are $1/3[1-210]$ a-type dislocations gliding on $(10-10)$ prismatic planes or $1/3[11-20]$ a-type dislocations gliding on $(1-100)$ prismatic planes. A crystal schematic was also added to Fig. 4 to better illustrate the observation direction.

Reviewer 1 new response: I have difficulty with the crystallography – in Fig. 4(c), the $[01-10]$ direction (from the g -vector) is nearly parallel to the slip bands. If the slip planes are as claimed, should that angle be closer to 30° ? Just a question of interest, I am sure that the authors have thought about this in more detail than have I!

Response to the reviewer's new response: We appreciate that the reviewer noticed this. It is correct that the $(01-10)$ g vector and the actual slip plane of the dislocations are 30 degrees from each other. But the imaging is conducted at a $[2-1-10]$ orientation, and the basal plane is perpendicular to the imaging plane, rendering any vectors in the basal planes projected in the same direction. That is why $(01-10)$ is parallel to the slip trace.

Our statement related to the assertion of cross-slip events is based on the observed jagged dislocations and the extensive dislocation entanglement. As the reviewer pointed out, this is indirect evidence, but it would be difficult to rationalize the jaggedness of these a-type dislocations in another way. We modified the manuscript to reflect the reasoning.

Reviewer 1 new response: Thanks.

1. Response to my second comment:

Response to reviewer's comment ii: As we detailed in the previous response comment, the primary slip plane of the planar slip bands in the alloys can be identified by doing trace analysis of the slip band. As shown in Fig. 4, due to the low symmetry of hcp systems, we can determine the Burgers vectors and the primary slip planes of the planar slip bands with the g - $3g$ weak-beam images taken with $g = (0002)$ and $g = (01-10)$ near the $[2-1-10]$ orientation. At this condition, the basal plane is perpendicular to the imaging plane, therefore the observed planar slip bands consist of prismatic slip. According to the invisibility condition, the possible Burgers vectors are $1/3[11-20]$ and $1/3[1-210]$. So, the dislocations in the slip bands shown in Fig. 4 are $1/3[1-210]$ a-type dislocations gliding on $(10-10)$ prismatic planes or $1/3[11-20]$ a-type dislocations gliding on $(1-100)$ prismatic planes.

Unfortunately, we did not locate suitable basal planar slip bands for imaging.

Reviewer 1 new response: Are you still claiming prismatic and basal slip?

Response to the reviewer's new response: Yes, we performed trace analysis that indicated the fraction of basal slip in Ti-6Al-0.10 and Ti-6Al-0.30 can reach $\sim 40\%$, in contrast to only $\sim 2\%$ of basal slip in Ti-0.30. This suggests that the Al addition can indeed significantly promote the basal slip. This is also in a good

agreement with the calculations results in literature, in which the difference in CRSS between basal and prismatic slip in hcp Ti can be greatly reduced with increasing Al content.

Response to my third comment:

Response to reviewer's comment iii: We appreciate the reviewer's comment on this discussion. It is true that we do not have direct evidence of Frank-Read sources generating dislocations. We have modified the manuscript in page 10 as follows.

More importantly, the nodes of dislocation tangles are consistent with the theory that they can act as Frank-Read sources to emit dislocation loops into the undeformed areas in-between these slip bands

Reviewer 1 new response: Good response.

Response to other comments:

iv. The authors refer to individual O interstitials lying on a slip plane as being obstacles to dislocation motion. Is this meant to be different from solid solution strengthening?

Response to reviewer's comment iv: We appreciate the reviewer's comment on this topic. This is indeed a critical question related to the oxygen sensitivity of titanium. Solid solution strengthening usually refers to the interaction of the strain fields of dislocations with the strain fields with the solute atoms. But the interactions we referred to in the manuscript is beyond solid solution strengthening, where the slip of dislocations will mechanically shuffle the crystallographic sites of oxygen atoms. Thus, this effect is different than conventional solid solution strengthening. We modified the manuscript in page 2 to reflect this point.

Reviewer 1 new response: Solid solution strengthening is a result of not only interaction with strain fields, but also local elastic modulus changes, and the possibility of promotion of some directional bonding which can result in strengthening. Any consideration of these effects?

Response to the reviewer's new response: In our shuffling model, the interstitial oxygen atoms can be shuffled by screw dislocations from stable octahedral site to metastable hexahedral site. This may result into some direction bonding or changes in local modulus, although it could be a different issue from the reviewer's comment. We will take your suggestion into our consideration in the future research.

v. The claim that this work points the way for low-cost Ti alloys is somewhat over the top. The applications of Ti alloys is mainly at ambient and intermediate temperatures, not under cryogenic conditions. I am not able to believe that the next generation of Ti alloys will all contain 6Al and 0.30O! I believe that this speculation should be removed from the paper – the content is already of very high quality, and this unsubstantiated claim detracts from this quality.

Response to reviewer's comment v: We agree with the reviewer, this is a fair comment. The current manuscript is primarily focused on the deformation mechanism of the featured alloys. Any engineering applications would need further investigations and consideration of other factors such as cost. Accordingly, we have modified the final sentence of the conclusion as follows:

“These discoveries revealed a novel strategy to mitigate the undesired sensitivity and embrittlement of interstitial species such as oxygen, which could lead to new compositions of Ti alloys with higher oxygen tolerance to reduce processing costs.”

Reviewer 1 new response: Thanks.

vi. Change the term, "g dot b" analysis to "diffraction contrast".

Response to reviewer’s comment vi: We appreciate the reviewer’s comment and have modified the terminology accordingly.

Reviewer 1 new response: Thanks.

Summary: I remain with my opinion that this is a good paper that should be published. I have raised questions in my responses to their responses, and I think it would be advantageous to the authors if they would take these comments into consideration. However, I do not need to see their responses before recommending acceptance of the paper.